# Nanoscale dynamics of synaptic vesicle trafficking and fusion at the presynaptic active zone

**Thirumalini Vaithianathan[1], Diane Henry[1], Wendy Akmentin[1], Gary Matthews[1,2]***

[1]Department of Neurobiology and Behavior, Stony Brook University, New York, United States; [2]Department of Ophthalmology, Stony Brook University, New York, United States

**Abstract** The cytomatrix at the active zone (CAZ) is a macromolecular complex that facilitates the supply of release-ready synaptic vesicles to support neurotransmitter release at synapses. To reveal the dynamics of this supply process in living synapses, we used super-resolution imaging to track single vesicles at voltage-clamped presynaptic terminals of retinal bipolar neurons, whose CAZ contains a specialized structure—the synaptic ribbon—that supports both fast, transient and slow, sustained modes of transmission. We find that the synaptic ribbon serves a dual function as a conduit for diffusion of synaptic vesicles and a platform for vesicles to fuse distal to the plasma membrane itself, via compound fusion. The combination of these functions allows the ribbon-type CAZ to achieve the continuous transmitter release required by synapses of neurons that carry tonic, graded visual signals in the retina.

*For correspondence: Gary.G. Matthews@stonybrook.edu

**Competing interests:** The authors declare that no competing interests exist.

## Introduction

Neurons communicate at synapses, where chemical neurotransmitter is released from the presynaptic terminal by fusion of transmitter-containing synaptic vesicles with the plasma membrane. A macromolecular complex, the cytomatrix at the active zone (CAZ; *Südhof, 2012*) is thought to capture and organize synaptic vesicles to support neurotransmitter release by presynaptic terminals (*Sigrist and Schmitz, 2011*; *Gundelfinger and Fijtova, 2012*; *Hallermann and Silver, 2013*), but the dynamics of this process at living synapses remain largely elusive. In order to follow the trafficking of vesicles in real time at the CAZ, methods are needed to label and image both vesicles and the active zone at the same time in living synapses. To label the living active zone, we used fluorescent Ribeye-binding peptide (RBP; *Zenisek et al., 2004*), which marks the synaptic ribbon found at the active zones of retinal bipolar cells (BPCs) and other neurons that release neurotransmitter continuously in response to graded changes in membrane potential (*Matthews and Fuchs, 2010*). However, monitoring the trafficking of tiny synaptic vesicles (30–50 nm in diameter) is a technical challenge, which we were able to meet by targeting a photoactivatable fluorescent protein to synaptic vesicles and using super-resolution photoactivated localization microscopy (*Betzig et al., 2006*) to track movements of single vesicles in voltage-clamped synaptic terminals. We also targeted the exocytosis reporter pHluorin (*Sankaranarayanan et al., 2000*) to synaptic vesicles (*Granseth et al., 2006*; *Voglmaier et al., 2006*) in order to detect vesicle fusion. These approaches then allowed us to analyze the trafficking and fusion of vesicles at the active zone of BPC ribbon synapses during ongoing neurotransmitter release.

Previously, studies of synaptic vesicle trafficking and fusion at ribbon synapses of BPCs have been carried out using total internal reflection fluorescence microscopy (TIRFM) to image single vesicles labeled with FM dye (*Zenisek et al., 2000, 2002*; *Holt et al., 2004*; *Midorikawa et al., 2007*;

**eLife digest** Neurons communicate with one another through junctions known as synapses. When a neuron is activated, it triggers the release of chemicals called neurotransmitters at the synapse, which bind to and activate neighbouring neurons. Neurons involved in vision, sound and balance contain "ribbon" synapses, which are able to release neurotransmitters steadily over longer periods of time than other types of synapse.

Neurotransmitters inside neurons are packaged into small structures called vesicles, which can then fuse with the cell's surface membrane to release the neurotransmitters from the cell. Unlike other types of synapse, ribbon synapses are able to release these vesicles in a continuous fashion. How vesicles move around at the synapses remains poorly understood because monitoring the vesicles in living cells is technically difficult and previous studies were limited to tracking vesicles in a small part of the synapse. Now, Vaithianathan et al. overcome these technical hurdles to follow the movement of vesicles across whole ribbon synapses in zebrafish eyes.

The experiments use fluorescent proteins to track the movement of the vesicles under a microscope. Vaithianathan et al. find that vesicles at ribbon synapses move very little when the neurons are not active. However, when the neurons are activated, the vesicles that are near the cell membrane fuse with it and release their neurotransmitters. Other vesicles that are further away from the membrane then move to fill in the spaces vacated by the fusing vesicles.

Further experiments show that some of the vesicles that are further away from the membrane can fuse with vesicles that have already released their neurotransmitter but remain in place at the membrane. This process – known as compound fusion – allows neurotransmitters to be released over a longer period of time by providing a path for vesicles to release neurotransmitters without having to approach the membrane first. The next challenge is to develop a computational model using the data from this study to better understand how ribbon synapses work.

*Zenisek, 2008*; *Joselevitch and Zenisek, 2009*). Some of the conclusions from this work are: 1) vesicles stably associate with ribbons in the absence of stimulation ('residents'), 2) these resident vesicles rapidly undergo exocytosis in response to depolarization, and 3) new vesicles ('newcomers') appear, move toward the membrane, and fuse during sustained depolarization. Although TIRFM is a powerful approach to study membrane-associated phenomena, it is limited to imaging labeled vesicles within ~100 nm of the plasma membrane (e.g., *Zenisek et al., 2000*), which is insufficient to provide coverage of all the ribbon-associated vesicles. The method also requires tight adherence of the plasma membrane to a planar substrate, which restricts observations to a small part of the terminal and eliminates the natural membrane curvature in that observable region. Furthermore, to our knowledge, only *Midorikawa et al. (2007)* and *Zenisek (2008)* have combined vesicle imaging with ribbon labeling, and then only in sequentially acquired images separated by some time, which were intended to test whether observed hotspots of fusion coincided with ribbons. Because of these limitations, we used two-color laser scanning methods that allowed single labeled vesicles to be observed throughout the full extent of the ribbon in voltage-clamped synaptic terminals, while the ribbon and cell border were imaged with a second fluorescent label. Since the positions of the ribbon and a labeled vesicle were known accurately, we were able to detect vesicle movements on the ribbon prior to fusion and determine where vesicles resided on the ribbon when they fused. Therefore, our experiments complement and significantly extend the previous studies based on TIRFM.

## Results

To track single vesicles, we targeted the photoactivatable fluorescent protein paRFP (*Subach et al., 2010*) to synaptic vesicles by fusing it to the C-terminus of the vesicle membrane protein Vglut1 (vesicular glutamate transporter 1). The resulting fusion protein was then inducibly expressed in transgenic zebrafish (*Figure 1*; *Figure 1—figure supplement 1*). Synaptic vesicles bearing Vglut1-paRFP participated normally in the vesicle cycle, since they were labeled by activity-dependent uptake of FM dye. By photoactivating Vglut1-paRFP at low density, we ensured that a field of view contained only a single fluorescent synaptic vesicle, which we then localized using high-resolution

confocal microscopy (*Figure 1A–F*; *Figure 1—figure supplement 2*). Both mobile and immobile vesicles were visible after photoactivation (*Video 1*), but the immobile vesicles were closely associated with ribbons (*Figure 1A,F*; *Video 2*), which were labeled with green-fluorescent RBP introduced via a whole-cell patch pipette. Therefore, we conclude that the ribbon-associated vesicles are immobile because they are tethered to or near the ribbon itself (*Figure 1F*). These immobile vesicles likely correspond to the 'resident' vesicles observed at BPC active zones using TIRFM (*Zenisek et al., 2000*; *Midorikawa et al., 2007*), except that our laser-scanning method detected vesicles located everywhere on the ribbon instead of only in a region <100 nm from the plasma membrane (the region marked with pink shading for illustrative purposes in *Figure 1F*).

Resident vesicles at ribbons move very little in the absence of stimulation (*Zenisek, 2008*), and we therefore used the temporal jitter in the observed position of paRFP-labeled vesicles relative to ribbons to estimate the spatial precision of the localization method under our imaging conditions. A series of x-y images was acquired while the synaptic terminal was voltage-clamped at -60 mV to prevent release, and after averaging over 4–10 frames to reduce noise, the positions of the paRFP spot and ribbon were determined by fitting 2D-Gaussians, as illustrated in *Figure 1E*. Vesicle position was then expressed relative to the center of the ribbon (i.e., the peak of the ribbon 2D-Gaussian) in order to control for small movements of the cell and/or ribbon during an experiment. The vesicle position estimated in this manner varied little during the 90-s imaging period (*Figure 1G–I*), within ranges that averaged 27 nm on the x-axis and 17 nm on the y-axis. Assuming that vesicles actually did not move during the imaging, we attribute the observed variation in estimated position to imaging noise that limits the spatial resolution. Therefore, genuine stimulus-induced vesicle movements on the ribbon of approximately a vesicle diameter would be resolvable by comparing averaged x-y images before and after stimulation.

## Vesicle turnover and replenishment on ribbons

What happens to the immobile ribbon-associated vesicles during transmitter release? To control release, isolated BPCs were voltage-clamped via a whole-cell pipette placed directly on the synaptic terminal (*Figure 1—figure supplement 1B*) and stimulated by brief depolarization from -60 to 0 mV to activate voltage-gated calcium channels (*Figure 2A*, inset), causing calcium influx that triggered synaptic vesicle fusion. BPCs exhibit two kinetically distinct components of transmitter release during depolarization (for zebrafish BPCs, see *Vaithianathan and Matthews, 2014*): a fast component depleted within 10 ms, and a slower component lasting hundreds of ms. We first determined how individual paRFP-labeled vesicles at the ribbon responded when the system was perturbed by releasing just the fast component, which is thought to represent the cohort of vesicles tethered to the ribbon nearest the plasma membrane (*Matthews and Fuchs, 2010*), where voltage-gated calcium channels reside.

On some trials after a 10-ms stimulus, a new stable paRFP spot appeared at the ribbon (*Figure 2A,C*), while on other trials, a previously present paRFP spot disappeared after the stimulus (*Figure 2B,D*), representing single-vesicle replenishment and loss, respectively. When the loci of gained and lost paRFP spots were determined as illustrated in *Figure 1* and plotted with respect to the center of the ribbon (*Figure 2E*), there was no difference in the distribution of appearances and disappearances along the axis parallel to the membrane (the y-axis; p = 0.4, Wilcoxon rank test). However, disappearances clustered significantly closer to the plasma membrane than appearances (*Figure 2E*) along the axis perpendicular to the membrane (the x-axis; $p<10^{-8}$, Wilcoxon rank test). This pattern revealed by single-vesicle imaging is consistent with selective fusion of membrane-proximal synaptic vesicles during brief depolarization, but preferential recruitment of new vesicles at the opposite pole of the ribbon to replace those lost, as suggested previously from population behavior of vesicles labeled with FM dye (*Vaithianathan and Matthews, 2014*). A simple interpretation is that loss of membrane-proximal vesicles creates the opportunity for vesicles remaining on the ribbon to rearrange and occupy the vacated region, opening up 'slots' for new vesicles at more distal positions on the ribbon.

## Vesicle redistribution on ribbons after stimulation

To determine whether vesicles remaining on the ribbon indeed changed position after depolarization, we measured the relative positions of single paRFP-labeled synaptic vesicles that were present

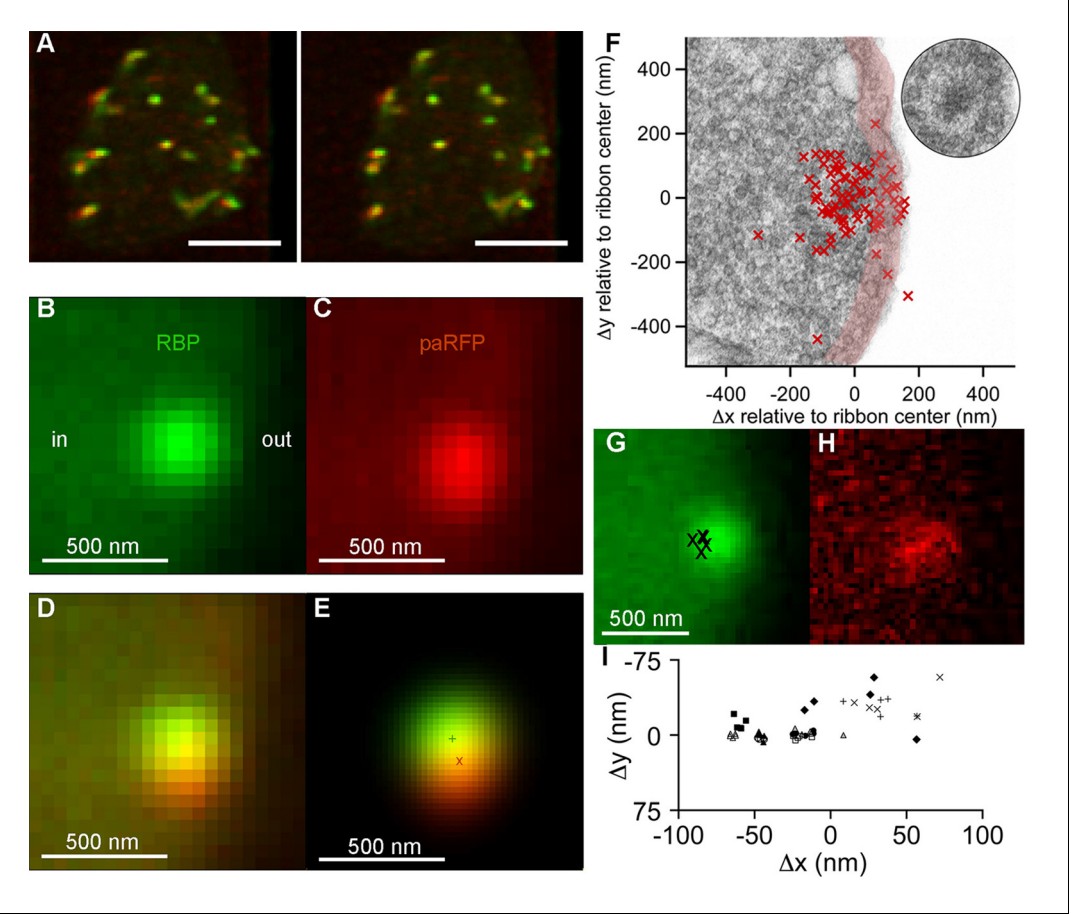

**Figure 1.** Synaptic vesicles labeled with Vglut1-paRFP stably associate with synaptic ribbons at the active zone. (**A**) Stereogram of a 3D reconstruction from a series of confocal optical sections through the synaptic terminal of a zebrafish BPC, showing photoactivated Vglut1-paRFP (red) near synaptic ribbons (green). A rotating view of the reconstruction is provided in *Video 2*. Scale bar = 2 μm. Details of recording and photoactivation are provided in *Figure 1—figure supplement 1*. (**B**) Close-up view of a synaptic ribbon labeled with green-fluorescent RBP, which also fills the cytoplasm. Outside of the cell is to the right, which is the standard orientation adopted for all images. Average of 16 frames. (**C**) View of the same region showing a single Vglut1-paRFP spot, averaged over 128 frames. (**D**) Superposition of images from **B** and **C**. (**E**) 2D Gaussians (*Figure 1—figure supplement 2*) fitted to RBP and paRFP fluorescence, with peaks marked by + and X respectively. (**F**) Positions of 81 paRFP spots (red Xs) relative to the center of the RBP-labeled ribbon, superimposed on an electron micrograph of a ribbon in a zebrafish BPC. The pink region depicts the ~100-nm region imaged in TIRFM experiments (except that the region would be planar in an actual TIRFM experiment). The inset reproduces the ribbon image without overlaid paRFP centroids. (**G**) Ribbon image with superimposed positions of a paRFP spot, shown by Xs, followed for 90 s without stimulation. Each position was determined by fitting a 2D Gaussian to a single averaged image of paRFP fluorescence like that shown in **H**. (**H**) An example of a single paRFP image, averaged over 10 frames, used to estimate paRFP positions shown in **G**. (**I**) Stability of paRFP location in the absence of stimulation for 10 experiments like that illustrated in G and H. Each symbol shows an individual paRFP location, and the different symbol types show results for a particular experiment.

The following figure supplements are available for figure 1:

**Figure supplement 1.** Overview of the experimental procedure.

**Figure supplement 2.** paRFP spots match the point-spread function of the microscope.

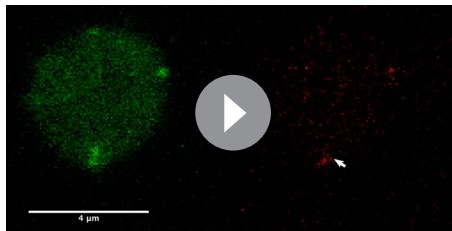

**Video 1.** Time-lapse movie of the synaptic terminal of a zebrafish bipolar neuron after photoactivation of Vglut1-paRFP. The arrow points to a stable vesicle present throughout, and mobile vesicles appear as transient flashes or streaks of red fluorescence during the x-y raster scans. Interval between frames: 3.77 s. Total duration of movie: 245 s. Each frame is an individual, unaveraged image.

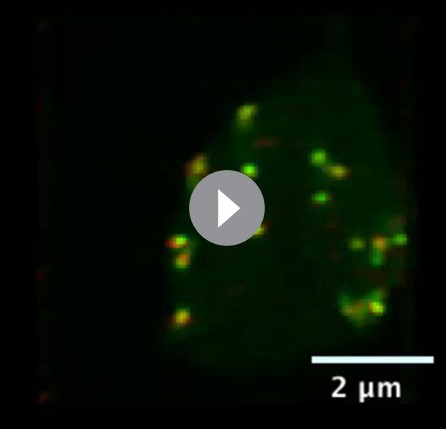

**Video 2.** 3D reconstruction of a living synaptic terminal of a zebrafish bipolar neuron from a z-axis series of confocal optical sections. Red spots are stable paRFP-labeled synaptic vesicles, and green spots are synaptic ribbons labeled with RBP.

on the ribbon both before and after 10-ms stimuli, or under the same conditions without stimulation. In the absence of stimulation, the position of labeled vesicles on the ribbon was stable, as shown by the histogram of displacement amplitude in *Figure 2F*. The great majority of such 'displacements' were less than a vesicle diameter and are therefore attributable to noise. However, after 10-ms stimulation, displacements were larger (*Figure 2F*; p≈0, Wilcoxon rank test), which is consistent with the prediction that the loss of ribbon-associated vesicles allows movement of remaining vesicles.

We next examined the direction of motion by plotting the vesicle displacement as a vector from the starting position. Without stimulation (*Figure 2G*), vector amplitudes were mostly within the range expected for imaging noise (93% were <50 nm; see inset in *Figure 2G* for an expanded view to better reveal the large number of small vectors), and the direction of the vector was equally likely to be toward or away from the membrane (positive and negative $\Delta\Delta x$, respectively, indicated by green and red vectors in *Figure 2G*). The average vector amplitude was therefore zero (black dot at the center of vector clusters). Rare larger excursions were also observed, which may reflect a low rate of genuine vesicle movement occurring spontaneously on the ribbon at the holding potential of -60 mV. After 10-ms stimuli (*Figure 2H*), vector amplitudes were significantly larger (77% were >50 nm; $p<10^{-6}$, Wilcoxon rank test), and there was a significant tendency for movement toward the membrane, with 70% of vectors having a positive $\Delta\Delta x$ (p=0.0008 that + and − $\Delta\Delta x$ were equally likely, by sign test). Amplitudes of vectors both toward the membrane (green vectors in *Figure 2H*) and away from the membrane (red vectors) were significantly larger after stimulation than in the absence of stimulation (+$\Delta\Delta x$: 89 ± 8 nm vs. 17 ± 1 nm, p≈0, Wilcoxon rank test; -$\Delta\Delta x$: 59 ± 6 nm vs. 16 ± 1 nm, $p<10^{-12}$, Wilcoxon rank test). Overall, the average vector after stimulation (black arrow, *Figure 2H*) was a displacement by 33 nm toward the membrane, or approximately one vesicle diameter. Since movements both toward and away from the plasma membrane were enhanced by stimulation, the results suggest a mechanism of passive diffusion of tethered vesicles along the ribbon, such as the recently proposed 'crowd-surfing' model (*Graydon et al., 2014*), rather than a directed motor. In such a passive mechanism, the net movement toward the membrane that we observed after stimulation is due to preferential release of vesicles near the membrane by 10-ms depolarization, creating the opportunity for more distal vesicles to diffuse into the vacated positions.

## Vesicle dynamics during the sustained component of release

In addition to the rapid burst of release at the onset of depolarization, ribbon synapses also release neurotransmitter continuously during sustained depolarization. To examine vesicle trafficking during this sustained component of release evoked by depolarization from -60 mV to -10 mV for >0.5 s, we measured fluorescence along a short scan-line perpendicular to the plasma membrane, which yielded x-t raster plots that were analyzed as shown in *Figure 3* to determine the x-axis position of

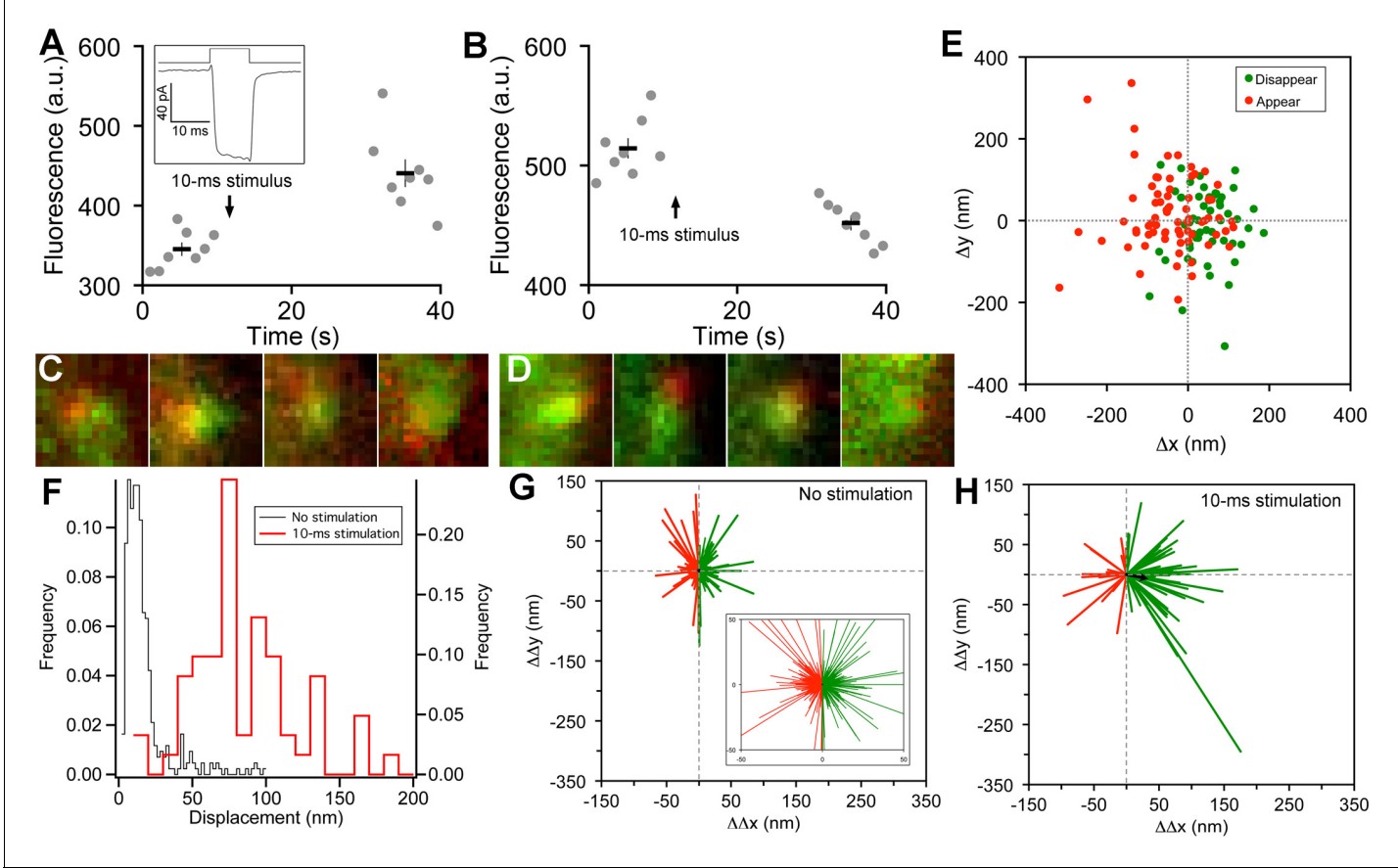

**Figure 2.** Vesicle release, replenishment, and movement elicited by brief depolarization. (**A**) Example of an increase in paRFP fluorescence after a 10-ms depolarization from -60 mV to 0 mV, which evoked the $Ca^{2+}$ current shown in the inset. Data points show fluorescence for individual frames, and the thick lines show the average ( ± sem) fluorescence over 8 frames before and after the stimulus. (**B**) An instance where depolarization caused a loss of paRFP fluorescence. (**C**) Representative examples of averaged images after depolarization for trials in which paRFP-labeled vesicles appeared post-stimulus. (**D**) Examples of averaged images before depolarization for trials in which paRFP-labeled vesicles disappeared post-stimulus. (**E**) Loci of paRFP-labeled vesicles with respect to the center of ribbon for 59 disappearances (green circles) and 71 appearances (red circles). (**F**) Histogram of displacement amplitude after 10-ms stimulation (red line) for 64 paRFP-labeled vesicles that were present before and after depolarization, compared with displacement histogram for 437 paRFP-labeled vesicles in the absence of stimulation (black line). Displacement = $(\Delta\Delta x^2 + \Delta\Delta y^2)^{0.5}$, where $\Delta\Delta x$ and $\Delta\Delta y$ are changes in paRFP locus relative to the center of the ribbon. (**G**) Displacement vectors without stimulation (N=437), shown on an expanded scale in the inset. The black dot in the center is the average vector. (**H**) Displacement vectors after 10-ms depolarization (N = 64).

paRFP spots relative to the synaptic ribbon and the plasma membrane (also see Image analysis section of Materials and methods). Although paRFP fluorescence was detectable in individual scan lines (*Figure 3E*), the noise level precluded accurate localization of the emitter by fitting the x-axis intensity profile of single lines. In the example shown in *Figure 3E*, for instance, the peak of the fitted Gaussian varied over a range of 170 nm in 16 consecutive individual lines. However, averaging across 4 scan lines reduced the range of variation of fitted positions to 20 nm, as shown in *Figure 3F*. As a result, we routinely averaged over 4–10 lines, depending on the noise level in a particular experiment, in order to achieve localization along the x-axis with a resolution of approximately one vesicle diameter. After such averaging, the position of a labeled vesicle in the absence of stimulation was stable for many seconds within a range of ± 25 nm (*Figure 3G,H*), which we take as the spatial precision of the line-scan method.

We next examined changes in paRFP-labeled vesicles at ribbons during activation of calcium current elicited by depolarizing voltage-clamp steps from -60 to -15 mV for 500–5000 ms. During sustained depolarization, previously stable paRFP spots often disappeared from the ribbon, either without a change in position before loss (e.g., *Figure 4A*) or after moving toward the membrane (e.

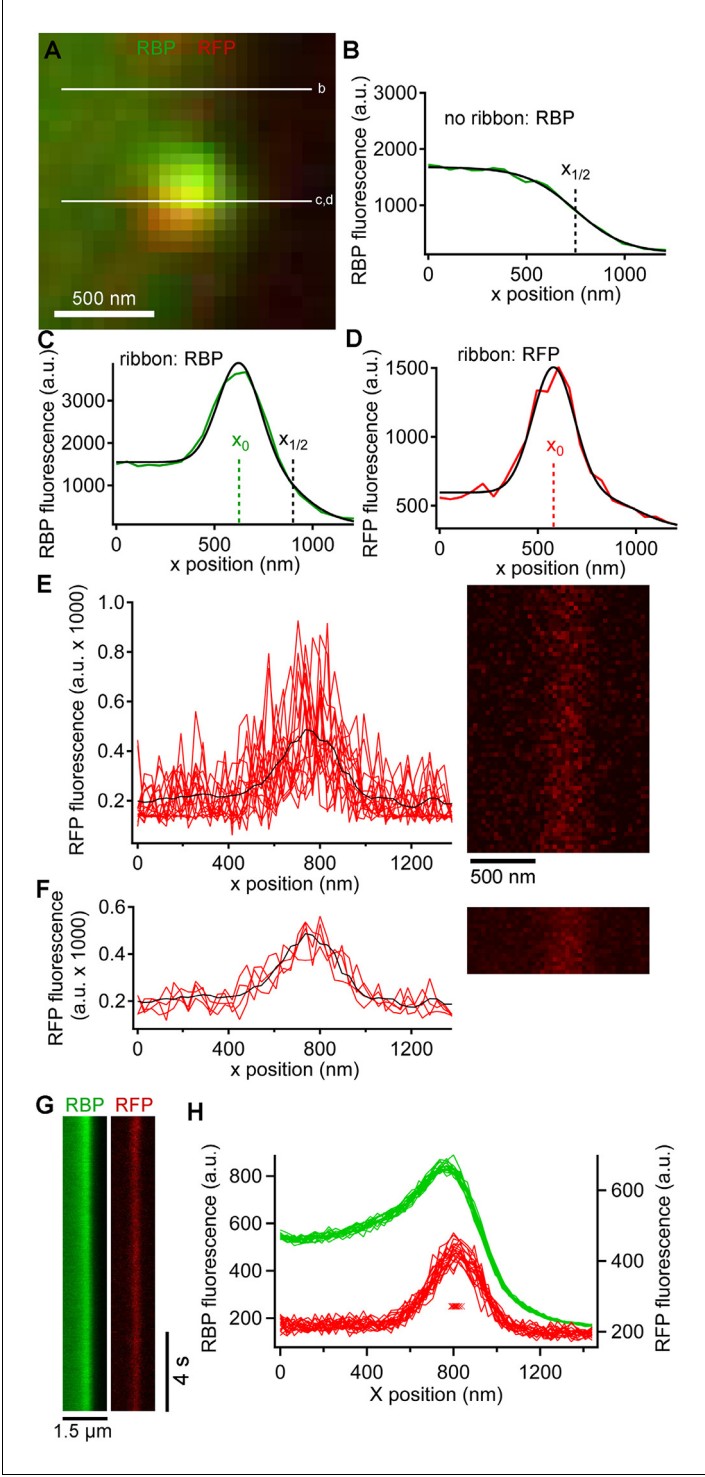

**Figure 3.** Generation and analysis of line scan data. (**A**) Scan lines were positioned perpendicular to the plasma membrane, extending from the intracellular side of the ribbon into the extracellular space. For illustrative purposes, a second scan line is shown at a non-ribbon location. Lower-case letters show line positions for panels **B–D**. (**B**) Intensity profile of green RBP fluorescence at the non-ribbon location. The parameter $x_{1/2}$ from the sigmoid fit (black line) is taken as the position of the membrane (see Materials and methods). (**C**) Intensity profile of green RBP fluorescence at the ribbon. The parameter $x_0$ is the peak of the Gaussian fit, giving the x-position of the center of the ribbon (see Materials and methods). (**D**) Intensity profile of paRFP fluorescence at the ribbon, with the x-position of the labeled vesicle given by $x_0$. (**E**) Example of x-t image of paRFP spot at a ribbon, *Figure 3 continued on next page*

*Figure 3 continued*

consisting of 64 line scans (right) taken over 472 ms. X-axis intensity profiles are plotted (left) for the first 16 line scans to illustrate the noise within and across individual line scans. The superimposed black line shows the average of all 64 lines. (F) Data from E were averaged over four lines to reduce noise and allow more precise localization of the paRFP spot along the x-axis. The intensity profiles show the first four of the temporally averaged line scans, with the average for all lines superimposed in black. (G) Example x-t images averaged over five line scans, showing a stable paRFP-labeled synaptic vesicle associated with a ribbon, which was labeled in green with RBP. Membrane potential was voltage-clamped at -60 mV. (H) Intensity profiles of RBP (green) and paRFP (red) taken in successive 406-ms intervals from the images in **G**. The red Xs show the position of $x_0$ for each paRFP trace, which varied over a range of 48 nm during the 15-s recording.

g., *Figure 4B,C*). Whether vesicles moved or not before disappearance depended on starting position with respect to the membrane (*Figure 4D*): vesicles nearer the plasma membrane (positive starting positions in *Figure 4D*) did not consistently move before disappearing, while distal vesicles (negative starting positions) moved toward the membrane before loss. Since membrane-proximal vesicles did not move while distal vesicles moved toward the ribbon center, the histogram of final paRFP positions just prior to disappearance was broad (*Figure 4E*) and covered the full range of ribbon-associated vesicle positions (cf., *Figure 1F*). During disappearance, paRFP fluorescence did not cease abruptly, as would occur for bleaching, but instead declined along a Gaussian time course, which is consistent with diffusion of the emitter away from the scanned region (*Figure 4—figure supplement 1*). It is uncertain at present whether this diffusion reflects movement of the Vglut1-paRFP protein itself, as would occur for example if the protein incorporated into the plasma membrane after vesicle fusion, or dissociation of an intact vesicle from the ribbon.

## Locus of vesicle fusion revealed by pHluorin reporter

Are vesicles that disappear from distal positions on the ribbon during sustained depolarization simply shed from the ribbon without participating in release, or do they somehow fuse and release their contents without approaching the plasma membrane? To detect the fusion of ribbon-associated vesicles, we generated transgenic zebrafish that express the exocytosis reporter SypHy (*Granseth et al., 2006*) (Synaptophysin-pHluorin fusion protein) or Vglut1-pHluorin (*Voglmaier et al., 2006*) under control of heat-shock promoter. Since pHluorin-fusion proteins mixed with pre-existing native vesicle proteins, pHluorin events were sparse during depolarization, allowing detection and localization of single events with respect to the ribbon, in the same manner as paRFP-labeled vesicles. For 2-color imaging, ribbons were labeled with deep-red CF633-RBP, which did not interfere with pHluorin fluorescence. Under these conditions, unitary pHluorin events representing vesicle fusion were observed in x-t raster plots during sustained depolarizing voltage-clamp steps from -60 to -15 mV for 500–5000 ms (*Figure 5A,B*), but not in the absence of stimulation. However, the signal-noise ratio for pHluorin fluorescence was poorer than for paRFP, and this necessitated averaging over a greater number of scan lines to achieve spatial resolution in the range of a vesicle diameter for pHluorin events, as illustrated in *Figure 5—figure supplement 1*. The fluorescence intensity profile of pHluorin events perpendicular to the membrane was then analyzed as described earlier for paRFP to determine the position of pHluorin events relative to the ribbon and the plasma membrane (*Figure 5C,D*). Events sometimes arose near the plasma membrane, suggesting fusion of membrane-proximal vesicles (e.g., *Figure 5C*), but events also commonly originated at the distal pole of the ribbon, at a distance from the plasma membrane (e.g., *Figure 5D*). The histogram of event positions relative to the center of the ribbon (*Figure 5E*) showed that pHluorin signals were broadly spread across the ribbon (*Figure 5E*) and were not confined to positions near the plasma membrane, as would be expected if vesicles must first approach the plasma membrane in order to fuse.

Because the z-axis thickness of our optical section was appreciable on the scale of our x-y resolution (see Materials and methods), one concern is that curvature of the plasma membrane in the z-axis could make fusions that actually occurred at the membrane near the top or bottom of the optical section appear to be nearer to the center of the ribbon. However, the majority of pHluorin events still arose at a substantial distance from the estimated membrane position, even after correcting for z-axis curvature (dashed arrow, *Figure 5E*). Another source of uncertainty in localizing pHluorin

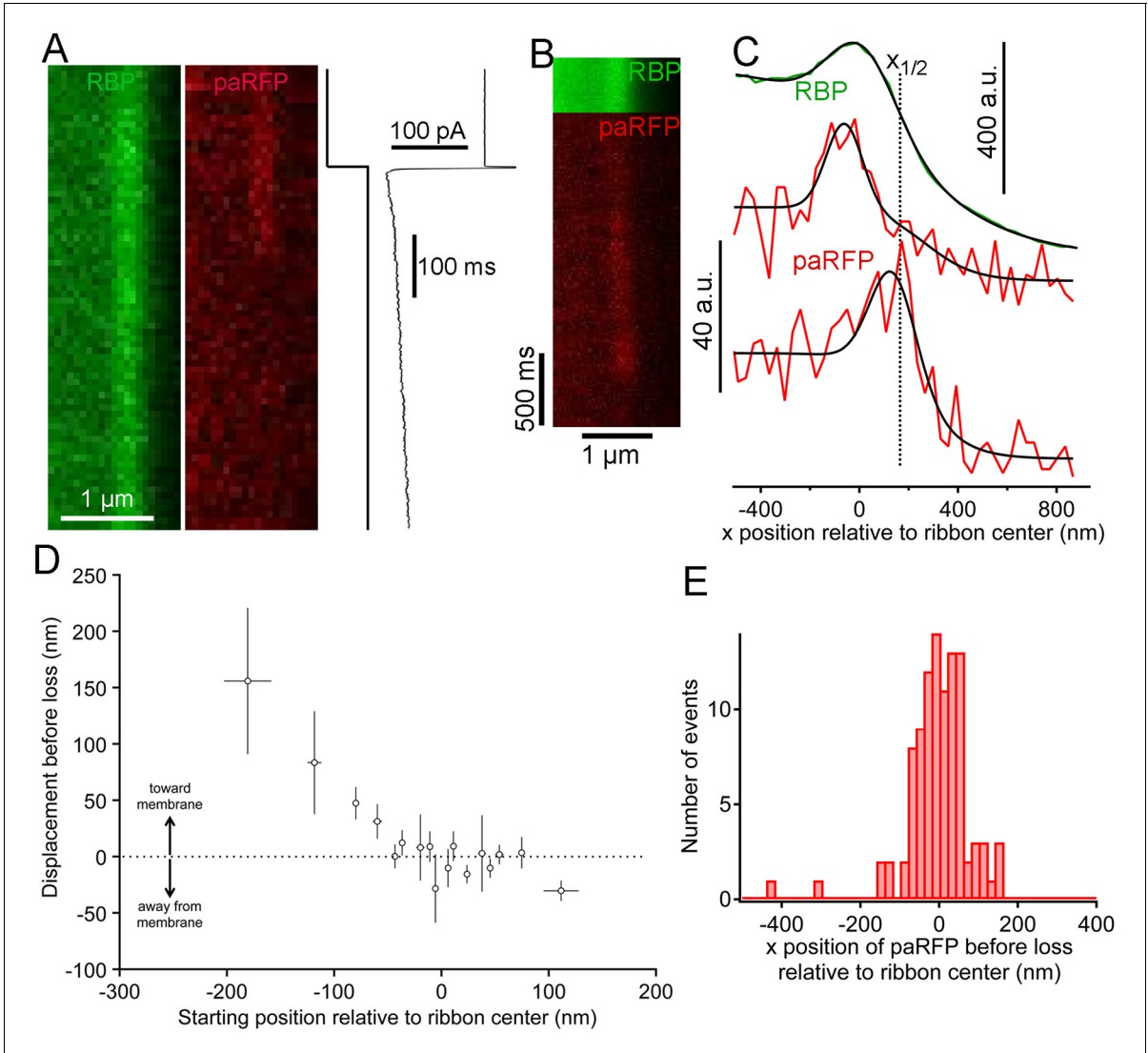

**Figure 4.** Movement and loss of vesicles during sustained depolarization. (A) Example x-t images in which a previously stable paRFP-labeled vesicle disappeared from view during sustained depolarization. The timing of depolarization and the evoked $Ca^{2+}$ current are shown to the right. Analysis of x-t line scan images is described in *Figure 3.* (B) Example showing a paRFP-labeled vesicle that appeared distal to the center of the ribbon during sustained depolarization, moved toward the membrane, and disappeared along a Gaussian time course (*Figure 4—figure supplement 1*). (C) Fluorescence intensity profiles along the x-axis for the example in panel B, showing ribbon position (green) and paRFP positions at appearance and disappearance (red). Black lines are fits described in *Figure 3.* The dotted line shows the estimated position of the plasma membrane estimated from $x_{1/2}$ obtained from the fit to RBP fluorescence. (D) Displacement amplitude for 88 paRFP-labeled vesicles along the x-axis prior to disappearance, as a function of initial starting position relative to the center of the ribbon. Positive and negative displacements are movements toward and away from the membrane, respectively. Open circles show the average of groups of five points binned by starting position. Error bars: ± 1 sem. Positive starting positions are nearer the membrane, and negative positions are farther away. (E) Histogram of final positions of paRFP-labeled vesicles along the x-axis just before disappearance.

The following figure supplement is available for figure 4:

**Figure supplement 1.** Fluorescence of an emitter declines along a Gaussian time course as the emitter moves away from the region of a line scan.

events could be changes in ribbon position along the z-axis within the optical section, since we normalized x-axis positions of pHluorin events with respect to the center of the ribbon in constructing the histogram shown in *Figure 5E*. Major errors seem unlikely to arise from such changes in ribbon position within the optical section, because light from the entire optical section contributes to the x-

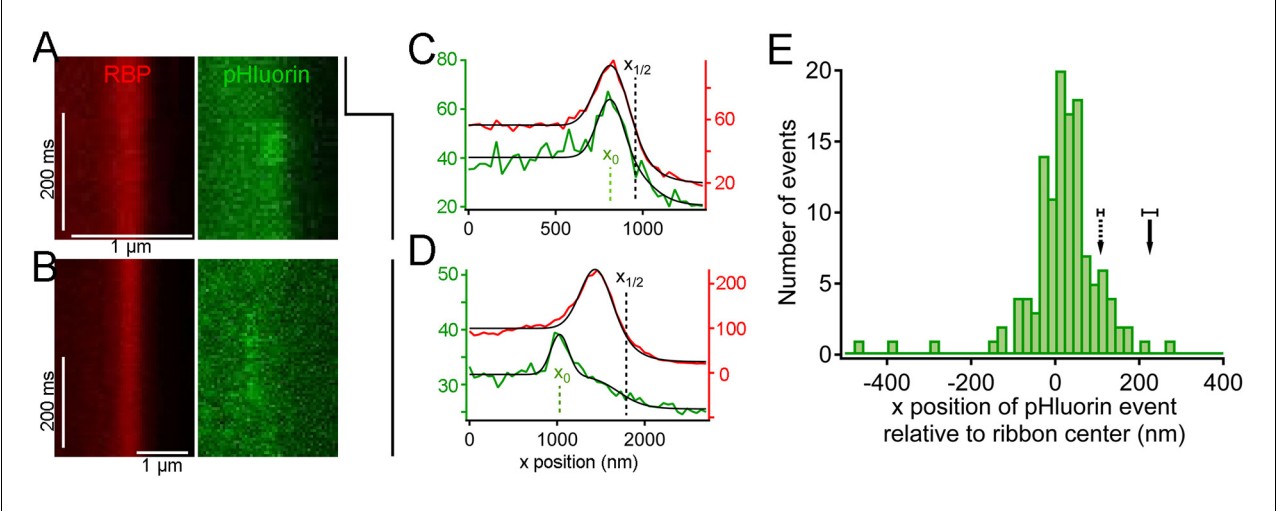

**Figure 5.** Synaptic vesicle exocytosis reported by pHluorin occurs at a distance from the plasma membrane. (A) Example of x-t images showing a pHluorin event from SypHy (green) at a ribbon shortly after onset of depolarization from -60 mV to -15 mV, shown by the trace to the right. Ribbon fluorescence from deep-red CF633-RBP is pseudocolored red. (B) Example of x-t images showing a pHluorin event from SypHy (green) on the membrane-distal side of a ribbon (red) in a different cell during depolarization to -15 mV. (C) (D) Fluorescence intensity profiles along the x-axis for RBP (red) and pHluorin (green) for the examples shown in A and B. Black lines are fits as described in *Figure 3A–D* and in Materials and methods. The x-axis position of the pHluorin event was taken to be the peak of the Gaussian from the fit ($x_0$, shown by the dotted green line), and the position of the plasma membrane was taken to be the parameter $x_{1/2}$ (dotted black line) from the fit to the fluorescence profile of the ribbon (red traces). (E) Histogram of the x-axis position of 127 pHluorin events from SypHy and Vglut1-pHluorin during sustained depolarization. The solid arrow shows the average relative position of the plasma membrane, estimated as described in Materials and methods (error bar: ± 1 sem). The dashed arrow shows the estimated position of the membrane at the top and bottom of the optical section, arrived at by assuming that membrane curvature observed along the y-axis for each x-y image also applied in the z-axis. Most of the pHluorin events fell outside the range of membrane positions between the two arrows.

The following figure supplements are available for figure 5:

**Figure supplement 1.** Noise level during pHluorin events.

**Figure supplement 2.** Variation in ribbon position in different focal planes.

axis intensity profile used to localize the ribbon in a given focal position. To estimate the extent of error from this source, we measured the change in x-axis position of ribbons as the focal plane was varied within a z-axis range from appearance to disappearance of the ribbon. As shown in *Figure 5— figure supplement 2*, the center of the Gaussian component of the fit to the RBP profile varied somewhat with focal position, especially near the upper and lower z-axis bounds, where fluorescence intensity of the ribbon was lower (e.g., sample traces in *Figure 5—figure supplement 2A,B*). However, the estimated x-axis position of the ribbon varied by less than ± 50 nm throughout the central part of the sampled z-axis range (*Figure 5—figure supplement 2C*). Therefore, the observed uncertainty in the x-axis position of ribbons within the optical section is insufficient to account for the wide range of pHluorin-event locations (*Figure 5E*).

We conclude that pHluorin events associated with ribbons arose both near the membrane and at more distal locations, at a substantial distance from the membrane. A simple explanation is that the distal pHluorin events represent fusion of synaptic vesicles with other ribbon-attached vesicles, which have themselves already undergone exocytosis and provide a path to the extracellular space. This model is also consistent with previous work using electron microscopy (*Matthews and Sterling, 2008*), which suggested the existence of such vesicle-vesicle compound fusion at ribbon synapses of goldfish BPCs.

## Discussion

In this study, we exploited the special features of ribbon synapses of retinal BPCs to observe the trafficking of single synaptic vesicles associated with the CAZ during ongoing transmitter release. Prior studies of synaptic vesicle trafficking at BPC ribbon synapses using TIRFM focused on a narrow region extending ~100 nm from the plasma membrane and revealed vesicle movements within this region, as well as vesicle exocytosis at the membrane (*Zenisek et al., 2000, 2002*; *Holt et al., 2004*; *Midorikawa et al., 2007*; *Zenisek, 2008*). By contrast, our imaging approaches allowed vesicle motion and fusion to be detected across the entire population of ribbon-associated vesicles, albeit at a spatial resolution insufficient for detection of small movements of less than a vesicle diameter, like those reported in TIRFM experiments (*Zenisek et al., 2000*). Therefore, our studies target a different set of questions and provide results that are complementary to the previous studies of synaptic vesicle dynamics at ribbon active zones.

Our results suggest a dual role for the synaptic ribbon in neurotransmitter release at the active zone, with the ribbon serving as both a conduit for diffusion of tethered synaptic vesicles and a platform for vesicles to fuse distal to the plasma membrane. A likely mechanism for such distal fusions is compound fusion of synaptic vesicles with other vesicles. Diffusion appeared to dominate for ribbon-associated vesicles farthest from the plasma membrane, which moved toward the center of the ribbon prior to loss during sustained depolarization. Synaptic vesicle fusion detected by pHluorin events occurred most frequently on the membrane-proximal half of the ribbon, perhaps because calcium levels are higher nearer the membrane during depolarization (*Zenisek et al., 2003*; *Francis et al., 2011*; *Vaithianathan and Matthews, 2014*). These fusion events include some that likely occurred directly at the plasma membrane, which possibly correspond to the fusions seen in TIRFM experiments. However, many pHluorin events arose at distances >100 nm from the estimated position of the plasma membrane, suggesting that compound fusion contributes significantly to ongoing transmitter release at ribbon synapses during sustained depolarization. Overall, our results set the stage for models of synaptic vesicle trafficking and fusion at the ribbon active zone that incorporate calcium signaling, the kinetics of tethered diffusion of vesicles, and vesicle-vesicle fusion at a distance from the plasma membrane.

## Materials and methods

### Animals

All animal procedures were in accord with NIH guidelines and followed protocol 247885 approved by the Institutional Animal Care and Use Committee of Stony Brook University. To generate transgenic zebrafish, transgenes were assembled using the Gateway-based Tol2kit (*Kwan et al., 2007*), by constructing appropriate middle-entry and 3'-entry vectors for combination with *hsp70* 5'-entry vector into a destination vector that included flanking Tol2 transposons and a reverse-orientation *cmlc2*:EGFP cassette for identifying transgenic embryos. To construct Vglut1 fused at the C-terminus with PATagRFP, cDNA for full-length Vglut1a (GenBank NM_001098755) was obtained by RT-PCR from zebrafish retinal total RNA and cloned into a middle-entry vector. PATagRFP cDNA was a gift from Dr. Vladimir Verkhusha (Albert Einstein College of Medicine) and was cloned into a 3'-entry vector. Zebrafish SypHy was generated as described (*Zhu et al., 2009*), using cDNA for Synaptophysin-b (GenBank NM_001030242) obtained by RT-PCR from zebrafish retinal total RNA and cDNA encoding super-ecliptic pHluorin (*Sankaranarayanan et al., 2000*) obtained by PCR from a SypHluorin vector (Addgene 37003; *Zhu et al., 2009*). The resulting zebrafish SypHy construct was then cloned into a middle-entry vector. Zebrafish VGlut1-pHluorin (*Voglmaier et al., 2006*) was produced in a similar manner. The assembled *hsp70* transgenes were injected into single-cell zebrafish embryos along with Tol2 mRNA, and transgenic embryos were selected at 24 hours postfertilization based on GFP fluorescence in the heart driven by *cmlc2*:EGFP. Transgenic fish were tested for germline transmission of the transgene at 3 months, and the positive progeny were then used to establish transgenic lines. For experiments, transgenic fish >3 months old of both sexes were exposed to 37° C water for 2 hr, kept overnight for protein expression, and used the next day.

## Patch-clamp recordings and imaging

Bipolar neurons were isolated from adult zebrafish retina as described previously (*Vaithianathan and Matthews, 2014*; *Vaithianathan et al., 2013*), and whole-cell recordings were made within 2 hr of dissociation, using a patch pipette placed directly on the synaptic terminal. Pipette and bath solutions were similar to those used for goldfish bipolar cells (*Heidelberger and Matthews, 1992*), except the pipette solution included 3 mM reduced glutathione as a free-radical scavenger and fluorescent RBP peptide to mark ribbons (*Zenisek et al., 2004*). Although it was concentrated at ribbons, fluorescent RBP also filled the entire cell, allowing the border of the synaptic terminal to be visualized with fluorescence imaging (see *Figure 3*). Voltage-clamp data were acquired using a HEKA EPC-9 amplifier controlled by PatchMaster software (HEKA, Lambrecht/Pfalz, Germany). Fluorescence images were acquired using Olympus FluoView software controlling an Olympus FV1000/IX-81 laser-scanning confocal microscope equipped with a 60X 1.42 NA oil-immersion objective. Separate xy-scanners allowed simultaneous imaging and photoactivation. The focal plane was carefully adjusted to bring labeled ribbons into sharp focus, avoiding the region of high curvature near the top of the terminal and the plane of adherence of the membrane to the glass coverslip at the bottom of the terminal. This procedure minimized curvature of the plasma membrane in the z-axis within the optical section and therefore facilitated localization of the plasma membrane with respect to the ribbon. Either x-y raster scans or x-t line scans were acquired of a zoomed region near a selected ribbon, depending on the experimental goals. Logic pulses exchanged between patch-clamp and imaging computers synchronized acquisition of electrophysiological and imaging data, and precise timing of imaging relative to voltage-clamp stimuli was established using PatchMaster to digitize horizontal-scan synch pulses from the imaging computer in parallel with the electrophysiological data. Electron microscopy of isolated bipolar cells was performed as described previously (*Matthews and Sterling, 2008*; *Vaithianathan et al., 2013*).

## Image analysis

FluoView images of x-y and x-t scans were imported into ImageJ for initial processing and analysis, and subsequent analysis was performed in IGOR Pro (Wavemetrics, Portland OR). For analysis of changes elicited by brief stimuli, 4–10 paRFP images were collected, followed by 10-ms depolarization or no stimulation, and the sequence was repeated at 18-s intervals. Images of the ribbon marked with RBP were taken either just before the series of RFP images, or in sequential alternation with each RFP image of the series. Images were then averaged for each repeat, and the positions of paRFP spots and the ribbon were determined from the peak of a 2D Gaussian fitted using IGOR Pro. Positions of paRFP spots were then expressed as $\Delta x$ and $\Delta y$ relative to the center of the ribbon for each repeat. Changes in relative RFP position across repeats were then calculated as $\Delta\Delta x$ and $\Delta\Delta y$. For analysis of changes during sustained stimuli, a line perpendicular to the membrane at a ribbon location was scanned at intervals of ~1.4 ms, with alternation between RBP and paRFP or pHluorin channels on successive lines, or on successive frames. The resulting x-t images were typically averaged in ImageJ over 2 pixels in the x-axis and 5–8 lines in the t-axis to reduce noise. Positions of the ribbon and paRFP spots along the scanned line were determined by fitting the x-axis intensity profile with the equation $f(x) = s(x) + g(x)$, where $s(x)$ is a sigmoid describing the transition from intracellular to extracellular background fluorescence at the edge of the cell, given by $s(x) = b-(c/(1-exp((x_{1/2}-x)/d)))$, and $g(x)$ is a Gaussian representing the fluorescence of RBP, paRFP, or pHluorin, given by $g(x) = a(exp(-(x-x_0)^2/w^2))$. The parameters $x_{1/2}$ and $x_0$ were taken as the x-axis positions of the plasma membrane and the fluorescence emitter, respectively. The parameter $b$ is intracellular background fluorescence, $c$ is extracellular background fluorescence, $d$ is the slope factor of the sigmoid, $a$ is the peak amplitude of emitter fluorescence, and $w$ is $\sqrt{2}*$ the standard deviation of the Gaussian. In practice, the latter parameters were highly constrained by the data or by the measured PSF, essentially leaving only $x_{1/2}$ and $x_0$ as free parameters in the fitting.

## Statistical methods

No statistical method was used to predetermine sample size. Non-parametric tests (Wilcoxon rank test; sign test) were used to determine statistical significance. Variance in estimates of the population mean is reported as $\pm$ sem.

## Acknowledgements

This work was supported by NIH grant EY003821 to G.M.

## Additional information

### Funding

| Funder | Grant reference number | Author |
|---|---|---|
| National Institutes of Health | R01EY003821 | Gary Matthews |

The funders had no role in study design, data collection and interpretation, or the decision to submit the work for publication.

### Author contributions

TV, DH, GM, Conception and design, Acquisition of data, Analysis and interpretation of data, Drafting or revising the article; WA, Acquisition of data, Analysis and interpretation of data, Drafting or revising the article

### Author ORCIDs

Gary Matthews, iD http://orcid.org/0000-0001-7915-9126

### Ethics

Animal experimentation: All animal procedures were in accord with NIH guidelines and followed protocol 247885 approved by the Institutional Animal Care and Use Committee of Stony Brook University.

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
