## [Decision Letter]

Thank you for submitting your work entitled "Nanoscale dynamics of synaptic vesicle trafficking and fusion at the presynaptic active zone" for consideration by *eLife*. Your article has been reviewed by two peer reviewers, one of whom is a member of our Board of Reviewing Editors, and the evaluation has been overseen by Gary Westbrook as the Senior Editor.

The reviewers have discussed the reviews with one another and the Reviewing Editor has drafted this decision to help you prepare a revised submission.

Both reviewers found that the approach taken is novel, clever and useful. They also found that your results provide a significant gain in a set of questions that has been debated for quite some time, and are therefore interesting to wider audience.

However, some of the conclusions require a better accounting of the apparent level of resolution obtained by the technique. In particular, the assessment of vesicle movement and localization of fusion events relative to the membrane are critically dependent upon knowing the true resolution limit of the technique. Since the resolution of single particle localization techniques is strictly a function of signal-to-noise, it is very likely different for each set of conditions and may even vary with protein expression level, degree to which out-of-focus fluorescent molecules are activated, other fluorophores in the cell and the imaging rate, among other things. One place where this is apparently addressed is in the first paragraph of the Results and Discussion, which states a resolution limit of 30 nm, which refers to Figure 1. Figure 1 plots the "stability" of the pa-RFP spot, but it really isn't clear what that shows (is that the SD in localization?, was the analysis/imaging performed exactly as it was for the stimulus experiments?). More specific comments are below.

A second criticism relates to that some of the results shown here mirror results previously obtained by others using TIRF. The previous work should therefore be better integrated in the Introduction and Discussion by the authors. For example, the idea the vesicle mobility is restricted by the ribbon was previously inferred using TIRF for goldfish bipolar cells. Similarly, the demonstration that membrane proximal vesicles make up the fast kinetic component of release is highly reminiscent of the 'resident' component in TIRF studies that make up the fast component of release and the directional movement toward the membrane in response to a stimulus is similar to the newcomers that make up the slower component. Along the same lines, the authors claim to observe "for the first time trafficking of individual vesicles associated with the CAZ during ongoing transmitter release", which is parsed out enough to possibly be technically correct, but is still misleading. The authors results, which follow proteins and provides information about vesicles at a depth from the membrane adds to our understanding of these previous studies. The authors really should discuss their results in the context of those previous findings.

The reviewers question whether the "movement" seen in absence of stimulation (Figure 2), are noise rather than "paRFP vesicles were equally likely to move toward or away from the membrane". Also, since the stability in the absence of a stimulus was used to define the resolution, why is this assumed to be vesicle movement (this seems like a circular argument)?

Related to this is the question of the size of the movement vector toward the membrane during stimulation. While it is clear that such net movement exists, the increase in the size of vectors away from the membrane are much less so. If the authors wish to conclude that there is significant movement of vesicles away from the membrane, then they should analyze those events independently.

Finally, the use pHluorin to determine whether compound fusion occurs in response to sustained depolarization raised some clarifications/concerns:

1) What kind of sustained depolarization was used?

2) What is the resolution of pHluorin localization and is it sufficient enough to conclude that fusion is happening away from the membrane?

3) The authors state that they correct for the curvature of the cell assuming the z-curvature matches the x-y curvature. This seems like a minor correction compared to the orientation of the ribbon. The methods state that they avoided active zones at the very top and bottom, but the ribbons need not be at the very top or bottom to have a skewed orientation. Were the ribbons all right at the equator of the terminal?

4) Can the authors rule out the possibility that some of their disappearance events are vesicles unbinding from the ribbon?

---

## [Author Response]

Both reviewers found that the approach taken is novel, clever and useful. They also found that your results provide a significant gain in a set of questions that has been debated for quite some time, and are therefore interesting to wider audience.

*However, some of the conclusions require a better accounting of the apparent level of resolution obtained by the technique. In particular, the assessment of vesicle movement and localization of fusion events relative to the membrane are critically dependent upon knowing the true resolution limit of the technique. Since the resolution of single particle localization techniques is strictly a function of signal-to-noise, it is very likely different for each set of conditions and may even vary with protein expression level, degree to which out-of-focus fluorescent molecules are activated, other fluorophores in the cell and the imaging rate, among other things. One place where this is apparently addressed is in the first paragraph of the Results and Discussion, which states a resolution limit of 30 nm, which refers to Figure 1. Figure 1 plots the "stability" of the pa-RFP spot, but it really isn't clear what that shows (is that the SD in localization?, was the analysis/imaging performed exactly as it was for the stimulus experiments?). More specific comments are below.*

It is correct that signal-noise ratio is the key, and we have added new figures and text to address this crucial point. A new paragraph (second paragraph in Results) is now devoted to Figure 1 to clarify what it shows and to specify how the measurements were made for x-y images. Similarly, the noise limitations of line-scan images are now addressed in a new Figure 3 and associated text (subsection “Vesicle dynamics during the sustained component of release”, first paragraph).

*A second criticism relates to that some of the results shown here mirror results previously obtained by others using TIRF. The previous work should therefore be better integrated in the Introduction and Discussion by the authors. For example, the idea the vesicle mobility is restricted by the ribbon was previously inferred using TIRF for goldfish bipolar cells. Similarly, the demonstration that membrane proximal vesicles make up the fast kinetic component of release is highly reminiscent of the 'resident' component in TIRF studies that make up the fast component of release and the directional movement toward the membrane in response to a stimulus is similar to the newcomers that make up the slower component. Along the same lines, the authors claim to observe "for the first time trafficking of individual vesicles associated with the CAZ during ongoing transmitter release", which is parsed out enough to possibly be technically correct, but is still misleading. The authors results, which follow proteins and provides information about vesicles at a depth from the membrane adds to our understanding of these previous studies. The authors really should discuss their results in the context of those previous findings.*

We have added a paragraph to the Introduction (second paragraph) in which we describe the prior results and state the motivation for our alternative approaches. We also added comparisons of our results with those from TIRFM in the Results section, first paragraph, and in both paragraphs of the Discussion.

*The reviewers question whether the "movement" seen in absence of stimulation (Figure 2), are noise rather than "paRFP vesicles were equally likely to move toward or away from the membrane". Also, since the stability in the absence of a stimulus was used to define the resolution, why is this assumed to be vesicle movement (this seems like a circular argument)?*

It was never our intention that the “movements” in the absence of stimulation should be viewed as real movements, and we in fact interpret them as noise, just as the reviewers state. We should have labeled them something like “apparent movement” and made it clear that they are likely due to imaging noise. We have now remedied this in the subsection “Vesicle redistribution on ribbons after stimulation “to clarify our intention.

*Related to this is the question of the size of the movement vector toward the membrane during stimulation. While it is clear that such net movement exists, the increase in the size of vectors away from the membrane are much less so. If the authors wish to conclude that there is significant movement of vesicles away from the membrane, then they should analyze those events independently.*

We have analyzed the movements toward and away from the membrane separately, as suggested. The added results and statistical significance levels are presented in the Results section, subsection “Vesicle redistribution on ribbons after stimulation”, second paragraph.

*Finally, the use pHluorin to determine whether compound fusion occurs in response to sustained depolarization raised some clarifications/concerns: 1) What kind of sustained depolarization was used?*

This is now specified in the first paragraph of the subsection “Locus of vesicle fusion revealed by pHluorin reporter”.

*2) What is the resolution of pHluorin localization and is it sufficient enough to conclude that fusion is happening away from the membrane?*

The spatial resolution of the pHluorin measurements is described in the first paragraph of the subsection “Locus of vesicle fusion revealed by pHluorin reporter” and in new Figure 5—figure supplement 1.

*3) The authors state that they correct for the curvature of the cell assuming the z-curvature matches the x-y curvature. This seems like a minor correction compared to the orientation of the ribbon. The methods state that they avoided active zones at the very top and bottom, but the ribbons need not be at the very top or bottom to have a skewed orientation. Were the ribbons all right at the equator of the terminal?*

The ribbons were all near the equator, but not necessarily right at it. The issue of ribbon orientation is now addressed extensively in a new paragraph (subsection “Locus of vesicle fusion revealed by pHluorin reporter”, second paragraph) that discusses sources of error in estimating the position of pHluorin events relative to the plasma membrane, and in a new supplemental figure (Figure 5—figure supplement 2).

4) Can the authors rule out the possibility that some of their disappearance events are vesicles unbinding from the ribbon?

We did not mean to exclude this possibility, which is now explicitly mentioned at the end of the subsection “Vesicle dynamics during the sustained component of release”.